# A Breakdown of Immune Tolerance in the Cerebellum

**DOI:** 10.3390/brainsci12030328

**Published:** 2022-02-28

**Authors:** Christiane S. Hampe, Hiroshi Mitoma

**Affiliations:** 1Department of Medicine, University of Washington, Seattle, WA 98195, USA; 2Department of Medical Education, Tokyo Medical University, Tokyo 160-0023, Japan; mitoma@tokyo-med.ac.jp

**Keywords:** cerebellum, autoimmunity, immune surveillance, ataxia, blood–CSF barrier, blood–brain barrier

## Abstract

Cerebellar dysfunction can be associated with ataxia, dysarthria, dysmetria, nystagmus and cognitive deficits. While cerebellar dysfunction can be caused by vascular, traumatic, metabolic, genetic, inflammatory, infectious, and neoplastic events, the cerebellum is also a frequent target of autoimmune attacks. The underlying cause for this vulnerability is unclear, but it may be a result of region-specific differences in blood–brain barrier permeability, the high concentration of neurons in the cerebellum and the presence of autoantigens on Purkinje cells. An autoimmune response targeting the cerebellum—or any structure in the CNS—is typically accompanied by an influx of peripheral immune cells to the brain. Under healthy conditions, the brain is protected from the periphery by the blood–brain barrier, blood–CSF barrier, and blood–leptomeningeal barrier. Entry of immune cells to the brain for immune surveillance occurs only at the blood-CSF barrier and is strictly controlled. A breakdown in the barrier permeability allows peripheral immune cells uncontrolled access to the CNS. Often—particularly in infectious diseases—the autoimmune response develops because of molecular mimicry between the trigger and a host protein. In this review, we discuss the immune surveillance of the CNS in health and disease and also discuss specific examples of autoimmunity affecting the cerebellum.

## 1. Immune Surveillance of the Central Nervous System (CNS)

The central nervous system (CNS) has long been viewed as immune-privileged, lacking interaction with the immune system. Today, it is understood that while there is considerably less interaction between the CNS and the immune system as compared to peripheral organs, the immune privilege is not absolute [1]. Instead, the brain undergoes constant immune surveillance by both microglia and peripheral immune cells [2,3].

Microglia are a central part of the innate immune system of the CNS [4]. As the resident immune cells in the brain, microglia constantly survey the CNS for signs of infection, ischemia, injury and disease [5].

While microglia reside in the brain and spinal cord, peripheral immune cells must transmigrate into the CNS either at the blood–brain barrier (BBB), the blood–CSF barrier, or the blood–leptomeningeal barrier (BLMB) [6,7].

Under physiological conditions, the BBB and the BLMB do not allow transmigration of peripheral immune cells (absolute barriers), and leukocytes enter the CNS via the BCSFB located at the choroid plexus (CP) (immune-regulatory barrier) [8].

### 1.1. Blood–CSF Barrier: Portal for Immune Surveillance during Physiological Conditions

The CP is located in each of the brain’s ventricles. Blood vessels in the CP are surrounded by layers of fenestrated endothelial cells lacking tight junctions that allow access for leukocytes to the stroma (Figure 1).

Trafficking of immune cells requires the traditional steps of capture, rolling, arrest, crawling and diapedesis [9]. The high blood flow rate at the CP (~5–10 times greater than in other tissues) contributes to the greater opportunity of circulating immune cells to migrate across the fenestrated capillaries into the stromal matrix [10].

The initial capture and rolling steps at the CP rely on the constitutive expression of P-selectin on the endothelial cells [11]. Recognition of P-selectin by its ligand expressed on leukocytes enables capture and rolling [11,12]. Once the cells have crossed the endothelial cell layer, they encounter different chemokines [13], and the expression of the corresponding chemokine receptor (CKR) on the immune cell determines whether the cell can transmigrate across the ependymal cell layer with its tight junctions [10]. The chemokine profile in the CSF is dominated by CCL2, CX3CL1, CCL3, CCL15 and CCL20 [10]. CCR6+ T cells may be attracted to the CP by CCL20 (CCR6’s only ligand) [14]. Indeed, CCR6-deficient T cells do not migrate across the BCSFB [14]. Finally, the tight junction proteins expressed in the ependymal cell layer differ from those present at the BBB and are more permissive for diapedesis [15]. Immune cells that cross the BCSFB enter the CSF of the respective ventricle [7], where they may engraft through interactions with VCAM-1 and ICAM-1 expressed on the apical side of the CP [16,17] or patrol the CNS via the CSF. For further reading and a more detailed discussion, we refer to the excellent review by Erickson and Banks [18].

The CSF totals approximately 140 mL in humans and is fully exchanged ~4 times every day. Under physiological conditions, the CSF contains up to 450,000 white blood cells, the majority (70–80%) being activated central memory T cells, with small contributions from B cells, NK cells, dendritic cells, mast cells, monocytes and granulocytes [10,19].

The CSF flows through the ventricles and through the subarachnoid spaces surrounding the brain and spinal cord. From there, it is either absorbed into the dural venous sinuses or is drained into the periarterial space, from which it travels through the brain tissues. It is then collected via the perivenous space into veins and is exposed to the lymphatic system [20]. CNS-derived antigens are collected in the CSF and presented to patrolling central memory T cells by dendritic cells and macrophages [21,22,23,24]. Importantly, under physiological conditions, immune cells remain in the CSF-drained perivascular, leptomeningeal or ventricular spaces and are separated from the CNS parenchyma by the glia limitans [24].

### 1.2. The Blood–Brain Barrier in Health and Disease

The BBB is composed of a monolayer of non-fenestrated endothelial cells linked by tight junctions, the basement membrane (consisting of laminins proteoglycans, fibronectin and type IV collagen), pericytes, microglia, and the glia limitans, consisting of astrocytic end-feet (Figure 2a). Under physiological conditions, immune cells do not cross the BBB.

#### 1.2.1. Inflammation and Breakdown of BBB Integrity

Adverse conditions such as acute brain injury, stroke, and infection lead to the activation of the microglia and subsequent release of inflammatory factors, such as cytokines, reactive oxygen species, and chemokines [25]. Initially, this immune response is limited to the CNS, but chronic inflammation results in the breakdown of the BBB [26,27] and the recruitment of peripheral immune cells for an inflammatory response [4,28].

The BBB can also be compromised by systemic inflammation when inflammatory molecules including prostaglandins, cytokines, chemokines, and nitric oxide cause the deterioration of the integrity of the BBB [29]. This affects all elements of the BBB: decreased expression of tight junction proteins [30], endothelial cell damage through apoptosis, membrane abnormalities, and mitochondrial damage [31], and the glia limitans begins to fail due to disruptions of astrocyte end-feet [32,33] (Figure 2b).

In addition to these disruptive changes, inflammation can also induce non-disruptive BBB changes [9,34]. Systemic cytokines can activate cytokine receptors expressed on the brain endothelium [35] and induce the expression of P-selectin (Figure 2b). Cytokines also stimulate endothelial production of chemokines such as CCL2 [36] and the expression of endothelial adhesion molecules, including ICAM-1 [37]. These changes in BBB permeability and/or cell surface proteins enable leukocytes to transmigrate across the BBB, beginning with tethering to the endothelium. Attached leukocytes crawl on the CNS microvessels to search for sites of diapedesis and cross the BBB via the paracellular or the transcellular route. Through this transendothelial passage, the cells enter the perivascular space. The second step of cell migration requires passage across the glia limitans to enter the brain parenchyma proper [1]. This step requires the degradation of tight junction proteins anchoring the astrocytic end-feet processes to the basement membrane [38].

Consequently, pathologic conditions, such as stress [39], infection and inflammation [40], traumatic brain injury or stroke [41], are accompanied by a drastic increase in numbers and variety of immune cells in the CNS.

As in peripheral autoimmune disorders, it is unclear what exactly causes the development of a CNS autoimmune response. Contributing factors include molecular mimicry, genetic susceptibility, the release of sequestered antigens, infections, exposure to chemicals or dietary components, and structural changes of proteins. Exposure of antigens that are usually sequestered in the CNS can cross the compromised BBB into the periphery [42,43] and trigger an autoimmune response.

Peripheral CNS-specific autoantibodies may gain access to brain tissue either by entering via a compromised BBB or non-specific transcytosis [44]. Likewise, the exposure of CNS-specific antigens to infiltrating lymphocytes can initiate an autoimmune response within the brain, which may be associated with intrathecal antibody production. Intrathecal autoantibody production is indicated by the presence of oligoclonal bands in the CSF, although these are not always present. Typically, these autoantibodies recognize different antigens and/or antigen epitopes than autoantibodies present in the periphery.

It is important to acknowledge that the BBB varies in cellular content and microvascular permeability in different brain regions [45]. These region-specific differences in BBB permeability may account for the higher vulnerability of distinct brain regions (e.g., the cerebellum) to autoimmune disorders.

#### 1.2.2. Region-Specific BBB Permeability

As discussed above, the BBB permeability is maintained by pericytes, perivascular macrophages, and astrocytes. However, the distribution, morphology and functionalities of these cells show brain region-specific differences, affecting the BBB permeability [45]. Astrocytes in different brain regions show distinct phenotypic and functional differences [46]. Pericytes are critical in the regulation of BBB permeability [47], and region-specific differences in pericyte coverage and functional heterogeneity contribute to differences in BBB permeability [48].

These regional differences in BBB permeability may affect the transmigration efficiency of immune cells and, thereby, CNS inflammation. Increased BBB permeability is observed in experimental allergic encephalomyelitis (EAE) mouse models and mice infected with rabies virus [49]. In both models, BBB permeability changes are observed mainly at the cerebellum and spinal cord and not the cerebral cortex, indicating that the BBB at the cerebellum is more vulnerable [50]. However, the distribution of CD4+ T cells is considerably different between these two models. In rabies-infected animals, CD4+ T cells are restricted to the vascular endothelium with no or few cells in the perivascular space or the parenchyma, respectively. In contrast, EAE mice show CD4+ T cell accumulation in the perivascular space and deep in the CNS parenchyma. These differences may be due to qualitative differences in BBB permeability. EAE mice show more extensive BBB permeability allowing large molecules (10 to 150 kDa) access to the CNS, while 150-kDa molecules do not cross the BBB during rabies infection, preventing antibodies from reaching infected CNS tissues.

Finally, experiments using in vitro cerebellar BBB models suggested that the BBB in the cerebellum is more vulnerable to inflammation, partly because of lower expression of BBB tight junction components claudin-1 and occludin and higher expression of VCAM-1 and ICAM [51].

## 2. Autoimmunity Specific to the Cerebellum

Different factors contribute to the high susceptibility of the cerebellum for autoimmune disorders (see above). Often, the presence of autoantibodies directed against cerebellar antigens aids in the diagnosis of autoimmune-mediated cerebellar ataxias (Table 1) [52]. In the following, we will discuss pathways leading to autoimmune disease involving both innate and adaptive immune responses. Overall, autoimmunity to the CNS (including the cerebellum) may be initiated with and without BBB or BCSFB compromise. The latter scenario can be caused by infectious pathogens that invade the CNS without damaging the BBB (see also below). However, in most cases, the integrity of the BBB or BCSFB is eventually compromised as a result of the ensuing inflammatory immune response.

### 2.1. Innate Immunity

Members of the innate immune response in the cerebellum include cerebellar microglia and astrocytes. Cerebellar microglia show a unique amoeboid morphology [56,57] and are in a hyper-alert immune state compared to microglia in other brain regions [58]. Gene expression studies revealed unique transcriptomes in cerebellar microglia, with enhanced expression of genes regulating immune alertness and energy metabolism [58]. The expression of these genes amplifies with age, which may contribute to the age-dependent onset of cerebellar ataxias. Furthermore, hyper-active microglia may further exacerbate inherited forms of ataxia [59], such as spinocerebellar ataxia type 6 (SCA6) [60], SCA1 [61], and Friedreich’s ataxia [62]. Cerebellar astrocytes fall into three main forms: white matter astrocytes, granular layer astrocytes, and Bergmann glia [63]. Bergmann glia are highly specialized astrocytes of the cerebellum and are critical for the proper function of Purkinje cells. Bergmann glia and Purkinje neurons are in close physical interaction, and the soma of Bergmann glia surround the cell bodies of Purkinje neurons [64]. Bergmann glia express high densities of glutamate transporters and thereby prevent excitotoxicity [65]. The reduction of glutamate transporters in Bergmann glia and subsequent Purkinje cell degeneration are associated with ataxia [61,66].

### 2.2. Adaptive Immune Response

As outlined above (Section 1.2.1), peripheral immune cells can gain access to the cerebellum and contribute to the development of cerebellar autoimmune diseases. The detection of B cells, autoantibodies and/or autoreactive T cells in patients presenting with cerebellar dysfunction point to the involvement of the adaptive immune response. For some of the disorders, the associated autoantigen has been identified, e.g., onconeuronal antigens in paraneoplastic cerebellar degeneration, while in other cases, the autoimmune target has yet to be associated with a specific etiology. The later cases are diagnosed as primary autoimmune cerebellar ataxia (PACA) until the etiology has been resolved [67].

However, autoantibodies and autoreactive T cells can be epiphenomenal, without a pathogenic role in the disease. A pathogenic role for autoantibodies and autoreactive T cells can be confirmed by adoptive transfer to a suitable animal model. The involved pathomechanism can be established in other experimental designs, including ex vivo assessment of toxicity to neurons, induction of cellular damage or immune activation, or interference with synaptic transmission using whole-cell recordings.

Generally, neurodegenerative diseases show an age-related progression, including many immune-mediated cerebellar ataxias that develop in middle age [68]. With age, both the adaptive [69] and the innate [70] arms of the immune system undergo changes. While T- and B-lymphocytes decrease in numbers and functionality [69], changes in the innate immune response result in an age-related low-grade inflammation [71]. These changes occur in parallel with a decline in the permeability of the BBB and the BCSFB. This is in part due to changes in cellular function and morphology [72] and expression levels of tight junction proteins [73]. This breakdown is region-dependent, occurring first in the hippocampus [74] and spreading to other brain regions. A compromised BBB and BCSFB will allow easier access of immune cells and proinflammatory cytokines to the CNS, accelerating the development of neurodegenerative diseases. Detailed discussions of the effect of age on BBB functions and the immune system are provided elsewhere [75].

## 3. Diversity in Cerebellar Autoimmunity

Immune-mediated cerebellar ataxias characteristically include divergent etiologies that reflect different autoimmune backgrounds, such as differences in autoimmune stimuli (transient or persistent) and induced effecter cells (innate immunity, adaptive immunity, or both).

In the following, we will discuss different triggers of autoimmune cerebellitis and their pathogeneses. Associated pathomechanisms and potential therapeutic approaches are included, which will shed light on still-unknown underlying mechanisms (Figure 3)

A recurring theme in cerebellar autoimmunity is molecular mimicry between the autoantigen and the triggering event. This is particularly evident in autoimmune responses triggered by infections.

### 3.1. Postinfectious Cerebellitis (PIC)

To enter the CNS, infectious pathogens first need to penetrate the BBB or BCSFB via transcellular or paracellular entry, or they may also ‘bypass’ the blood–CNS barrier by migrating within nerves or being transported along axons.

Transcellular penetration occurs either by receptor-mediated uptake or pinocytosis without disruption of cellular barriers. Viruses such as tick-borne encephalitis virus frequently use this pathway [76]. During paracellular entry, the pathogen transmigrates between cells. This pathway is either used when bone-marrow-derived cells are infected by the virus and enter via the BCSFB route (HIV [77], Ebola virus [78]), or pathogens disrupt tight junctions, thereby increasing BBB permeability (Influenza A [79]). Other pathogens migrate within nerves to the CNS (poliovirus [80]). Once in the CNS, the pathogen induces an inflammatory response, which may weaken the BBB integrity and thereby allow immune cells access to the CNS.

Viruses that specifically target the cerebellum include non-polio enterovirus [81] and West Nile virus [82]. Infectious agents associated with cerebellar autoimmune disorders include varicella-zoster virus [83,84], Epstein–Barr virus [85], *Mycoplasma pneumoniae* [86], rotavirus, mumps, rubella, and influenza [87,88]. While several associated autoantigens have been identified (centrosomes, myelin-associated glycoprotein, neurons, and triosephosphate isomerase), a pathogenic role for these autoantibodies remains to be established [83,84,85,86,89].

It remains unclear whether the autoantibodies are generated in the periphery as part of the systemic immune response to the pathogen and cross into the cerebellum during inflammation-mediated higher permeability of the BBB (outside-in), or whether the pathogen initiates the autoimmune response within the CNS (inside-out). In most cases, the disease resolves on its own, supporting a scenario where autoantibodies cross into the CNS without intrathecal autoantibody production. The CNS symptoms resolve after the infection is over and autoantibodies no longer have access to the CNS.

Pathomechanism: Autoimmune cerebellitis caused by *Mycoplasma pneumoniae* or *Campylobacter jejuni* are prime examples of autoimmunity developing through molecular mimicry. *M. pneumoniae* usually targets the respiratory system, causing lung infections, but can also directly infect the CNS [90]. In this case, neurological symptoms may arise within 7 days of the initial respiratory infection [91]. Cerebellitis arising ≥ 8 days after the initial infection is indicative of indirect damage, possibly brought on by autoantibodies [92]. At this time, *M. pneumonia* DNA levels in the CSF are low or absent, indicating the absence of active bacteria in the CNS. While antibodies directed to *M. pneumoniae* proteins are essential and sufficient to clear the pulmonary infection [93], antibodies directed against *M. pneumoniae* glycolipids are associated with the development of cerebellitis. These glycolipids exhibit homology with mammalian myelin glycolipid galactocerebroside (GalC), and antibodies directed against *M. pneumoniae* glycolipids crossreact with GalC [94]. Notably, in the absence of neuropathy, anti-GalC IgG is not observed during *M. pneumoniae* infection [93]. No passive transfer studies with anti-glycolipid antibodies have been performed to date to confirm the pathogenicity of the antibodies.

Molecular mimicry also underlies the pathogenesis of Miller Fisher syndrome (MFS). Characteristically, patients with MFS present with acute ophthalmoplegia, areflexia, and ataxia. The disease is often preceded by infections with *Camphylobacter jejuni* or Haemophilus influenzae [95]. The majority (70–90%) of patients test positive for autoantibodies directed against ganglioside GQ1b [96,97], and the antibody titer is correlated with the disease course [96]. These autoantibodies appear to arise as a result of molecular mimicry, with lipo-oliogosaccharides isolated from *C**. jejuni* or H. influenzae mimicking GQ1b [98,99,100]. The vast majority of MFS patients lack pathological manifestations of the cerebellum as assayed via MRI [101]. Instead of a central cause of ataxia, a sensory origin has been proposed [102], where autoantibodies bind to GQ1b expressed on muscle spindles [103]. This would also explain why MFS patients recover without residual deficits [104]. The pathogenicity of autoantibodies has not yet been confirmed in passive transfer studies.

Increasing number of cases with cerebellar ataxia in patients with COVID-19 are being reported, and in some cases, an autoimmune etiology has been proposed [105,106,107,108,109]. Affected patients tested positive for autoantibodies against NMDAR [105], GAD [106], CASPR2 [109], and anti-myelin oligodendrocyte glycoprotein (MOG) [108]. The pathomechanism by which SARS-CoV-2 infection triggers an autoimmune response in the cerebellum is not yet understood.

Treatment: In most cases, PIC is self-limiting [110], and close observation without medication is recommended. Should CA persist or progress, immunotherapy is recommended [111]. Plasma exchange has been successfully used for EBV-associated CA [112]. Given the recent nature of the disease, no standard treatment of COVID-19-associated CA has been developed; however, immunotherapy and/or steroid therapy have shown beneficial outcomes [113,114].

### 3.2. Paraneoplastic Cerebellar Degeneration (PCD)

Other cerebellar autoimmune diseases are observed in association with paraneoplastic neurologic disorders (PNS). PNS are characterized by an immune response directed against neuronal antigens that are expressed by tumor cells and neurons (onconeural antigens). The subsequent neurological symptoms are not caused by the tumor itself, but rather by the immune system’s response to cancer.

Paraneoplastic cerebellar degeneration (PCD) is one of the most common forms of PNS. Neurological symptoms include truncal and appendicular ataxia, dysarthria and nystagmus caused by the loss of cerebellar Purkinje cells [115,116]. These symptoms often precede the tumor diagnosis by months or even years [116,117].

The patients’ CSF often shows mild lymphocytosis, oligoclonal bands, and elevated total protein levels, indicative of the presence of immune cells and antibodies [115,118]. A large number of PCD-associated onconeural antigens have been identified, and the respective autoantibodies are associated with specific tumors (Table 1) [119]. Two of the best-described PCD autoantibodies target the cytoplasmic protein Yo and the voltage-gated calcium channel (VGCC).

#### 3.2.1. Anti-Yo Autoantibody-Associated PCD

Anti-Yo autoantibodies are typically found in female patients with breast and ovarian cancers [115,120,121], and less frequently in patients with endometrial, digestive or lung cancer [117,122,123,124]. The autoantibodies are directed against two proteins, CDR2 (cerebellar degeneration-related 2) and its paralog, CDR2L. CDR2L is the major autoantigen in PCD [125], and under physiological conditions, its expression is limited to cerebellar Purkinje cells, brain stem neurons and the testes [126,127]; however, little is known about its neuronal function.

While CDR2 and CDR2L share 45% of their sequences [126], their functions differ. CDR2L interacts with ribosomal proteins with functions in protein synthesis, while CDR2 binds to nuclear speckles and is involved in mRNA maturation [128]. Ectopic expression of CDR2 and CDR2L on ovarian tumor cells [129] may trigger the formation of autoantibodies [130] and autoreactive T cells [131], especially when accompanied by mutations in CDR2 genes [132].

Pathomechanism: Anti-Yo antibody-associated PCD is accompanied by cerebellar atrophy caused by extensive loss of cerebellar Purkinje neurons [133,134]. The cell loss is associated with inflammatory infiltrates present in the cerebellum composed of T and B cells, plasma cells and macrophages/microglial cells [132,134]. Involvement of both CDR2-specific CD8+ T cells [135] and/or autoantibodies [136] in Purkinje cell death has been proposed, but the debate regarding the pathomechanisms involved continues.

Autoantibody-mediated cell death is supported by studies showing antibody uptake by Purkinje cells [137,138,139,140] and the demonstration of cell death after uptake of CDR2 antibodies [136].

As in other autoimmune cerebellar diseases [53,141,142], anti-Yo autoantibodies dysregulate cellular calcium homeostasis, eventually leading to cell death [143]. Experiments in cerebellar organotypic slice cultures of rat brains confirmed uptake of anti-Yo antibodies by Purkinje cells and loss of calbindin D_28K_ [143,144]. Calbindin is a major calcium-binding protein and acts as a calcium buffer, and calbindin D_28K_ depletion has been reported in PCD [145], Parkinsons’s disease [146] and Alzheimer’s disease [147]. Anti-Yo antibodies thus interfere with intracellular calcium homeostasis, causing mitochondrial calcium overload and increased reactive oxygen species production, likely resulting in Purkinje cell apoptosis.

Other studies suggest that anti-Yo autoantibodies interfere with CDR2’s interaction with c-Myc and subsequent disruption of c-Myc cytoplasmic pathways, leading to accelerated neuronal apoptosis [148]. However, passive transfer experiments with CDR2 antibodies injected intraventricularly or into the brain parenchyma have not resulted in significant Purkinje cell death [137,149].

A T-cell mediated cell death is suggested by the high frequency of Yo-specific CD8+ T cells in PCD patients in some [150], but not all [151], studies. Some studies also detected cytotoxic T cells in the CSF [152] and cerebellum of patients [153]. As Purkinje cells express MHC class I, they could potentially present CDR2 peptides and thereby be recognized and subsequently destroyed by CDR2-specific CD8+ T cells [135]. However, the transfer of Yo-specific T cells into *scid* mice did not result in significant Purkinje cell death [149].

The currently most widely accepted pathogenetic scenario is that the ectopic expression of CDR2 and CDR2L antigens on tumor tissue results in an immune response, with the subsequent emergence of CDR2-reactive T cells and B cells in the CNS. After reaching the cerebellum, autoantibodies and/or autoreactive T cells cause damage of Purkinje cells [152], and autoantibody production continues intrathecally [154,155].

#### 3.2.2. VGCC-Associated PCD

A clearer picture of autoantibody-mediated ataxia in PCD has emerged for autoantibodies directed against the P/Q-type voltage-gated calcium channel (VGCC). While these autoantibodies are mainly found in patients with Lambert–Eaton myasthenic syndrome (LEMS) [156], they may also be present in patients with paraneoplastic and non-paraneoplastic cerebellar degeneration [157,158,159].

VGCC translate membrane depolarization to the increased influx of Ca^2+,^ supporting neurotransmitter release from the axon terminal [160]. The pathogenesis of VGCC autoantibodies is supported by the induction of ataxic symptoms in mice that received intrathecal administration of VGCC-Ab positive sera [161].

Pathomechanism: Acutely, VGCC-Ab may inhibit the channel function with a decrease in Ca2+ influx, leading to impaired synaptic transmission [160]. Longer exposure to VGCC-Ab may lead to channel internalization and loss of Purkinje cells, as observed in autopsy studies [158,162].

Treatment options for PCD: Treatment of associated malignancies are of paramount importance and may also alleviate paraneoplastic symptoms by removing the antigen-driving tissue. This treatment is followed by immunotherapies (corticosteroids, IVIg, plasma exchange, immunosuppressants, and rituximab, alone or in combination). The overall prognosis of PCDs is relatively poor, often because of metastasis [133], and improvement of neurological symptoms is rare [133]. However, a few cases of non-paraneoplastic CD showed improvement with rituximab and IVIg treatment [163,164]. Treatment is likely to be effective only if administered in the absence of irreversible damage of Purkinje cells [165].

While onconeural antibodies are often detected in patients with cerebellar ataxia prior to the detection of associated tumors, the opposite situation can also arise when onconeural antibodies are detected in patients with malignancies in the absence of cerebellar degeneration [166]. In rare cases, activation of the innate immune response by intravesical BCG treatment or a tick bite can trigger cerebellitis in cancer patients with preexisting systemic onconeural autoantibodies. The increase in systemic cytokine and chemokine levels may cause a compromise in the BBB integrity, allowing onconeural autoantibodies and associated immune cells access to the CNS [167].

For other autoimmune targets and pathomechanisms involved in PCDs, please see the excellent review by Loehrer et al. in this Special Issue [54].

### 3.3. Gluten Ataxia (GA)

Gluten ataxia (GA) is manifested as sporadic cerebellar ataxia associated with gluten sensitivity [168,169]. The reported prevalence ranges between 0–6%, depending on the respective study [170]. GA can also occur in the absence of intestinal symptoms [171]. Patients with gluten ataxia often have oligoclonal bands in their CSF, evidence of perivascular inflammation in the cerebellum, and autoantibodies reacting with Purkinje cells [172].

Patients with GA can show signs of cerebellar atrophy, which may be irreversible [173]. The presence of lymphocyte infiltrates in the cerebellum supports the involvement of the immune system in its pathogenesis [174].

Pathomechanism: Cerebellar tissue of patients often shows lymphocytic infiltrates consisting of T cells, B cells and macrophages [169]. Notably, cerebellar atrophy accompanied by cytotoxic T-cells, but in the absence of B cells of plasma cells, has been observed in the autopsy of a patient, suggesting the involvement of cellular autoimmunity [174]. Other studies point to a pathogenic role of autoantibodies against tissue transglutaminases TG2 and TG6 and anti-gliadin antibodies [118,175,176,177]. TG6 is primarily expressed in the brain, with weak expression on Purkinje cells [178], and cerebellar IgA deposits that contained TG6-Ab have been identified in postmortem tissue from patients with GA [179]. Development of ataxia in mice after intraventricular administration of the autoantibodies further support a pathogenic role of the autoantibodies [180].

Treatment: Adherence to a strict gluten-free diet (GFD) has been reported to result in clinical improvement in some [171,181] but not in other studies [182]. This heterogeneity may be due to the assessment of GFD adherence and progression of neuronal loss. It is recommended that adherence to a GFD should be strictly monitored to restrict antigens that can trigger autoimmune reactions.

### 3.4. GAD65Ab-Associated Cerebellar Ataxia

In other cases of autoimmune disorders affecting the cerebellum, the trigger of the autoimmune response is unknown; these are primary, or idiopathic, autoimmune disorders. While many of these diseases can also be triggered by the above-discussed pathways, in the absence of a clinical history of infection, cancer or CNS injury, an idiopathic etiology is assumed.

High titers of autoantibodies directed to the smaller isoform of glutamate decarboxylase (GAD65) can be found in some patients with idiopathic cerebellar ataxia [183,184,185].

Pathomechanism: A pathogenic role for GAD65Ab was supported in vivo by passive transfer experiments when intracerebellar administration of GAD65Ab in rats and mice induced ataxic symptoms [118,119,120,121], and in vitro by depressed GABA release in cerebellar brain slices incubated with GAD65Ab [140,186,187,188].

GAD65 has two roles in GABAergic neurotransmission, namely the synthesis of GABA from glutamate and the shuttling of GABAergic synaptic vesicles to the synaptic cleft [189,190]. GAD65Ab interferes with both GAD65 enzyme activity [191] and the association of GAD65 with the cytosolic side of synaptic vesicles [189,190]. This results in a decrease in vesicular GABA contents and inhibition of GABAergic vesicle transport to the synapsis [140,186,187,188]. Under normal conditions, the released GABA spills over to the neighboring excitatory synaptic terminals and inhibits presynaptic glutamate release through GABA receptors. However, inhibition of GABA release interferes with this mechanism, resulting in the elevation of glutamate release [186]. Taken together, GAD65Ab elicits marked imbalances between GABA and glutamate. This is further accelerated through the involvement of microglia and astrocytes [185]. Microglia activated by excessive glutamate levels can secrete various cytokines, which facilitate glutamate release presumably through xc(-) on microglia, and suppress the uptake of glutamate through excitatory amino acid transporters (EAAT) on astrocytes [192,193]. Thus, the neuroinflammation-induced chain reactions accelerate the imbalance, leading to profound excitotoxicity. In agreement with this notion, the cerebellar neurons are completely lost in patients with advanced-stage GAD65Ab-associated cerebellar ataxia [194].

While the cytoplasmic location of GAD65 appears to contradict a direct involvement of GAD65Ab in the pathogenesis, GAD65Ab are internalized in cultured AF5 cells [140,195], and anti-GAD65 monoclonal antibodies were observed in CA1 interneurons and Purkinje neurons shortly after injection in the medial septum/diagonal band and ipsilateral interpositus nucleus, respectively [196], suggesting that GAD65Ab can gain access to their cytoplasmic target [137,138,139,140], similar to the cellular uptake of Yo autoantibodies discussed above.

A pathogenic role of GAD65-specific T cells in neurological disorders has been suggested by studies demonstrating that administration of monoclonal GAD65-specific CD4+ T cells induced neuronal death and ataxia in mice [197]. However, T cell infiltration was not observed in the cerebellum, calling T cells’ contribution to cerebellar dysfunction into question.

Treatment: Patients respond well to immunotherapy aimed at the removal or reduction of GAD65Ab, either by plasma exchange or rituximab [165,198].

### 3.5. mGluR1-Associated Cerebellar Ataxia

Another example for idiopathic autoimmune cerebellar ataxia is associated with autoantibodies directed against the metabotropic glutamate receptor (mGluR1). mGluR1 is a cell surface receptor mainly expressed on Purkinje cells. Its activation through binding by L-glutamate initiates calcium signaling in Purkinje cells [199]. Autoantibodies directed against mGluR1 can be found in patients with ataxia with [200,201] and without evidence of an accompanying malignancy [202].

Passive transfer experiments in mice strongly indicated a direct pathogenic effect of the antibody with the development of ataxia [200]. Experiments in cerebellar mouse slice cultures demonstrated that the application of mGluR1-Ab reduced the basal activity of Purkinje cells [203].

Pathomechanism: The location of the autoantibody epitope at the N-terminal, ligand-binding extracellular domain of the receptor suggests that the immediate impact of the antibodies on Purkinje cells is likely mediated by blockade of the receptor and associated reduction in the excitability of PC [200,201,203]. Long-term exposure likely causes cerebellar atrophy with PC loss, as demonstrated in patients [201,203].

Treatment: Treatment with IVIg and/or plasma exchange may result in improvement of cerebellar ataxia in some, but not all, patients [200,201,202]. These differences may be due to individual stages of cerebellar atrophy or intrathecal antibody production.

## 4. Conclusions

Autoimmune cerebellar diseases are diverse regarding their antigenic targets, clinical phenotypes, pathogenic mechanisms, and initiating triggers. To enter the CNS, immune cells must transmigrate the BBB, the BLMB, or the BCSFB. Under physiological conditions, the main point of entry is at the BCSFB, located at the choroid plexus. Entrance is restricted to CD4+ memory T cells, which survey the CNS via the CSF. Inflammatory immune responses trigger a breakdown of the BBB permeability, allowing immune cells to enter the CNS while allowing neuronal antigens to enter the periphery. However, BBB permeability is brain-region specific. The BBB at the cerebellum appears to be more permeable compared to other brain regions.

Autoimmunity at the cerebellum can be triggered by a number of factors, including infectious pathogens, paraneoplastic cerebellar degenerations, and gluten sensitivity, but may also arise without any clear etiology.

The mechanism of the autoimmune pathogenesis may involve both autoantibodies and autoreactive T cells or the innate immune response.

## Figures and Tables

**Figure 1 brainsci-12-00328-f001:**
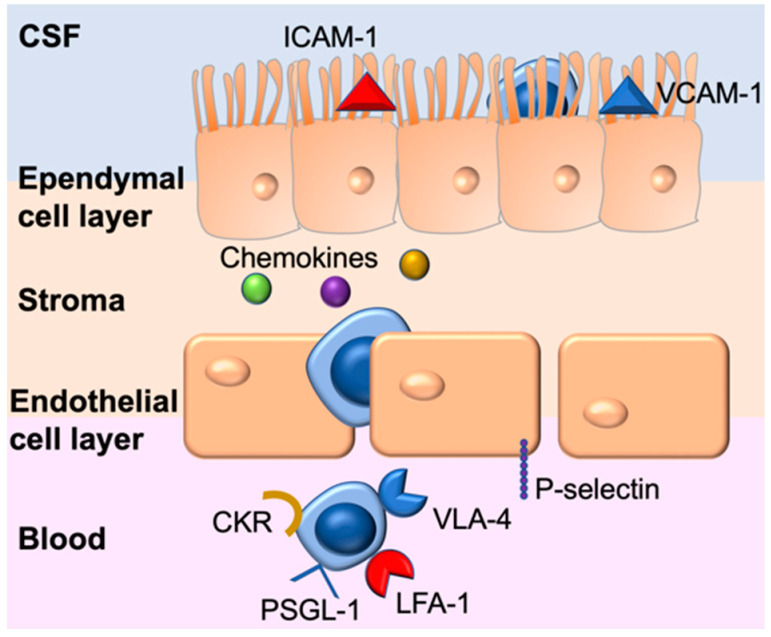
Lymphocyte migration across the BCSFB. Lymphocytes expressing PSGL-1 interact with P-selectin constitutively expressed on the endothelial cell layer of the CP. In the stroma, lymphocytes expressing chemokine receptors (CKR) matching the chemokines present at the CP are attracted to migrate across the ependymal cell layer. In the CSF of the CP, lymphocytes can either attach via interactions between integrins LFA-1 and VLA-4 with the respective cell adhesion molecules ICAM-1 and VCAM-1, or remain in the CSF.

**Figure 2 brainsci-12-00328-f002:**
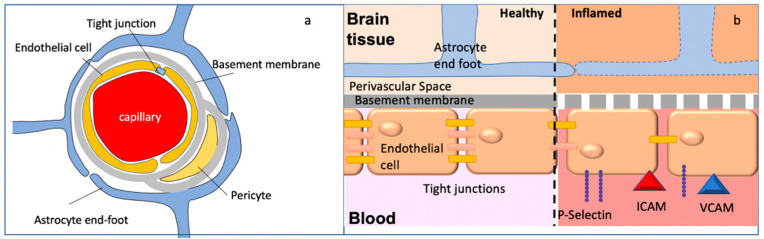
Blood–brain barrier. (**a**) Cellular components of the BBB. For details, see text. (**b**) Breakdown of BBB integrity. Inflammatory conditions induce the expression of P-selectin on the endothelial cells of the BBB, allowing PSGL-positive lymphocytes to tether. Tight junction proteins decrease in expression, and the basement membrane breaks down. Finally, the breakdown of the astrocyte end-feet allows access to the brain parenchyma.

**Figure 3 brainsci-12-00328-f003:**
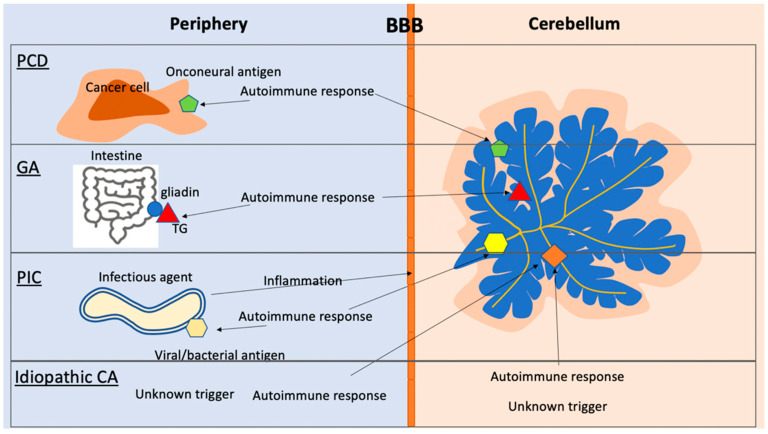
Autoimmune cerebellar diseases. PCD: the onconeural antigen expressed on the cancer cells triggers an autoimmune response, which can recognize the same antigen in the cerebellum. PIC: The infectious agent evokes an immune response to an antigen that bears similarity to a neural antigen (mimicry). The infection also causes an inflammatory response, which compromises the BBB, allowing antibodies and immune cells access to the CNS. GA: Autoimmune response to the TG-gliadin complex results in TG-specific autoantibodies. TG autoantibodies can recognize brain-specific TGs. Idiopathic CA: the triggering event resulting in the autoimmune response is unclear. The trigger may occur in the periphery or in the CNS.

**Table 1 brainsci-12-00328-t001:** Autoantibodies associated with autoimmune CA (compiled from [52,53,54,55]).

Autoantigen	Clinical Presentation	Association with Cancer
AP3B2	CA, peripheral neuropathy	Unclear
Ca/ARHGAP26	CA	Rare
CARP VIII	CA	Breast cancer, ovarian cancer, colorectal cancer, SCLC
Caspr2	CA, LE, Morvan syndrome	SCLC, thymoma
CV2/CRMP5	CA	SCLC, thymoma
GAD65	LE, CA	SCLC, neuroendocrine tumors, thymoma
GFAP	CA	Thymoma, ovarian, prostate, breast cancer
GluRdelta 2	CA, LE	Unclear
GlyR	CA, encephalitis	Thymoma, breast cancer, Hodgkin’s lymphoma, SCLC
Homer3	CA, encephalitis	SCLC
Hu	CA, LE	SCLC
IgLON5	CA	None
ITPR1	CA, encephalitis,	Breast cancer
Yo	CA	Breast, ovarian cancer
KLHL11	Brainstem syndrome, CA	Testicular cancer
LGl1	CA, encephalitis	Thymoma, neuroendocrine tumors
Ma2	LE, CA, brainstem encephalitis	Testis and lung cancer
mGluR1	CA	Hodgkin’s lymphoma
mGluR2	CA	unclear
Neuro chondrin	CA	None
PCA-2	Limbic/brainstem encephalitis, LEMS, CA	SCLC
Ri	CA	Breast cancer
Septin 5	CA	None
SEZ6L2	CA	Unclear
TG 2, 6	Gluten ataxia	None
Tr/DNER	CA, LE	Hodgkin’s lymphoma
TRIM 46, 9 & 67	CA	SCLC
VGCC	CA, LEMS	SCLC

CA: cerebellar ataxia, LE: limbic encephalitis, LEMS: Lambert–Eaton myasthenic syndrome, SCLC: small-cell lung cancer.

## Data Availability

Not applicable.

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
