# Peer review of "A Breakdown of Immune Tolerance in the Cerebellum"

_brainsci, 2022, doi:10.3390/brainsci12030328_

Round 1

Reviewer 1 Report

In this review paper, the authors summarize the available knowledge concerning the peculiar sensitivity of the cerebellum to autoimmune attacks, leading to different forms of cerebellar dysfunctions. This is a very interesting topic which deserves attention and is clearly dissected by the authors. First, detailed descriptions of how the immune surveillance occurs physiologically in the brain and is broken in pathology are provided and represent a very well-conceived introduction to the subsequent cerebellar focus. Later, the authors comment on how the cerebellum can be an extremely vulnerable target of autoimmunity, listing and explaining the distinct forms so far recognized of cerebellar dysfunctions due to this issue. Overall, this review paper is very well conceived, clearly written and provides a comprehensive view of the addressed topic, thoroughly dissecting the etiology, evolution and clinical signs of each autoimmune cerebellitis.

Very few minor points should be addressed to further ameliorate this excellent paper.

  1. The regional specificity of BBB permeability is a very interesting topic. Another point to discuss could be whether aging also affects this permeability and whether this results in a further increased cerebellar susceptibility later in life, leading to other forms (if known) of cerebellitis more typical of the mature age.
  2. Each section dedicated to the distinct forms of cerebellitis is very well discussed. Nevertheless, a comment on any available therapeutic approach to treat or even alleviate the symptoms should be added to provide a full picture of each disease.
  3. A brief summary of the different forms of autoimmune cerebellar pathologies and relative information should be added in form of a figure or table in the end of the manuscript.

Other minor comments:

  1. The numbering of the paragraphs from n. 2.2 should be reconsidered. Section n. 2.3 sounds more like a prosecution of the previous one, and should be numbered 2.3.1 or even incorporated in it.
  2. Similarly, the distinct forms of cerebellitis definitely deserve another dedicated section (i.e. 3, further subdivided into 3.1, 3.2 etc), possibly preceded by a short introduction like what is currently written in lines 199-201 but separated from the rest.

Author Response

  1. The regional specificity of BBB permeability is a very interesting topic. Another point to discuss could be whether aging also affects this permeability and whether this results in a further increased cerebellar susceptibility later in life, leading to other forms (if known) of cerebellitis more typical of the mature age.

We included a section discussing the effect of age on BBB permeability. While immune-mediated cerebellar ataxias usually occur during middle age, no investigations regarding the association of age on BBB permeability at the cerebellum were found.

  1. Each section dedicated to the distinct forms of cerebellitis is very well discussed. Nevertheless, a comment on any available therapeutic approach to treat or even alleviate the symptoms should be added to provide a full picture of each disease.

We added a section regarding treatment for each form of cerebellitis.

  1. A brief summary of the different forms of autoimmune cerebellar pathologies and relative information should be added in form of a figure or table in the end of the manuscript.

A figure depicting the different forms and pathogeneses of autoimmune cerebellar diseases has been added.

Other minor comments:

  1. The numbering of the paragraphs from n. 2.2 should be reconsidered. Section n. 2.3 sounds more like a prosecution of the previous one, and should be numbered 2.3.1 or even incorporated in it.

We amended the text accordingly.

  1. Similarly, the distinct forms of cerebellitis definitely deserve another dedicated section (i.e. 3, further subdivided into 3.1, 3.2 etc), possibly preceded by a short introduction like what is currently written in lines 199-201 but separated from the rest.

We agree with the reviewer. We added an introduction to cerebellar autoimmunity under a separate section (3).

Reviewer 2 Report

The main issue about the present Review is its general organisation. The content of the different main chapters and their subchapters is incoherent, lacking an overview that balances the choice of the given information.

Chapter 1. Immune surveillance of the Central Nervous System (CNS)

The entire chapter and sub-chapters deal with the general topic as indicated in the title, there are few figures that help the reader and the register is simple but clear. This style clashes with the next part, that instead offers a more specialised perspective. Furthermore, the connections with the cerebellum, which seems to be the main interest, are only sporadic.  For example: a premise is that the cerebellum is particularly vulnerable and suggests that this may be due to "region-specific differences in blood brain barrier permeability" but the chapter 1.2.1 highlights this peculiarity of the cerebellum only marginally.

Chapter 2. Autoimmunity specific to the cerebellum

The style of this chapter changes abruptly, offering very precise experimental and methodological details in some parts but not in others, without making explicit the rationale for this choice.

Section 2.1: The title of this section is “Innate immunity”, definitely a broad concept that then, in the text, is proposed from a very narrow perspective. The Authors mention the role of microglia in ataxia and a very comprehensive review [46], but then they elaborate this wide topic mentioning only two very specific examples. For example: the SCA6 ataxia is only one of the PolyQ ataxias where an hyperactivation of the microglia might be involved. Depending on the aim of the Authors they should treat this topic either in a more detailed way or with a more general, but complete, approach.

Section 2.2. Adaptive immune response

Among the different types of immune-mediated cerebellar ataxias only the Primary Autoimmune Cerebellar Ataxia deserves a specific citation at the end of the first paragraph [50]. Then, the text treats specific technical issues until the end of the section. This is the first time in the manuscript in which diagnostic methods are the primary interest of the Authors, the result could be disorienting for the reader.

Section 2.3: Confirmation of a pathogenic role for autoantibodies and autoreactive T cells

This seems to introduce the following sections, it should not be proposed at the same “Level” (2.3, 2.4, 2.5.) of the next chapters.

Section 3.2.1 Anti-Yo autoantibody associated PCD

Section 3.2.1 arrives after section 2.5. Starting from this section the Authors introduces a new formal convention, the sub-section "Pathomechanism" that was never seen before.

After section 3.2.2. there is a new 2.5 section.

Minor issues

Abstract: Line 7-9: It is hard to summarise the role of cerebellum and the neurological implications in two sentences, the mentioned “predictive fashion” does not describe the much more complex and still debated role of the cerebellum, the Authors should try to stick with a stricter neurophysiological interpretation.

Line 46-48: The mechanisms underlying the cell trafficking is not mentioned while it could better introduce the reader into the topic.

Figure 2 (b): The disruption of the BBB organisation in inflammatory conditions is not immediately apparent. The differences between the “Healthy” and “Inflamed” side should be better highlighted in the figure.

Line 109-110: In this sentence, the impact of stress on BBB integrity is compared to the effects of infections and brain injuries. Such a strong statement would benefit of a specific citation.

Line 112-113: Since the autoimmune response seems to be one of the main topic in this review it may deserve more attention. The passage from an immune to an autoimmune response is described without mentioning the mechanisms underlying this transition.

The readability of the first chapter benefits from the presence of schematic figures. The second part of the Manuscript deals with even more complex mechanisms but lacks figure, its readability would greatly improve if the Authors added graphic schemes here too.

Author Response

Chapter 1. Immune surveillance of the Central Nervous System (CNS)

The entire chapter and sub-chapters deal with the general topic as indicated in the title, there are few figures that help the reader and the register is simple but clear. This style clashes with the next part, that instead offers a more specialised perspective. Furthermore, the connections with the cerebellum, which seems to be the main interest, are only sporadic.  For example: a premise is that the cerebellum is particularly vulnerable and suggests that this may be due to "region-specific differences in blood brain barrier permeability" but the chapter 1.2.1 highlights this peculiarity of the cerebellum only marginally.

We have added/moved text to the end of 1.2.1 to better connect between the subchapters.

Chapter 2. Autoimmunity specific to the cerebellum

The style of this chapter changes abruptly, offering very precise experimental and methodological details in some parts but not in others, without making explicit the rationale for this choice.

An introductory paragraph has been added.

Section 2.1: The title of this section is “Innate immunity”, definitely a broad concept that then, in the text, is proposed from a very narrow perspective. The Authors mention the role of microglia in ataxia and a very comprehensive review [46], but then they elaborate this wide topic mentioning only two very specific examples. For example: the SCA6 ataxia is only one of the PolyQ ataxias where an hyperactivation of the microglia might be involved. Depending on the aim of the Authors they should treat this topic either in a more detailed way or with a more general, but complete, approach.

We have rewritten the section and taken a more general approach, including also contributions of the cerebellar astrocytes in cerebellar autoimmune disorders.

Section 2.2. Adaptive immune response

Among the different types of immune-mediated cerebellar ataxias only the Primary Autoimmune Cerebellar Ataxia deserves a specific citation at the end of the first paragraph [50]. Then, the text treats specific technical issues until the end of the section. This is the first time in the manuscript in which diagnostic methods are the primary interest of the Authors, the result could be disorienting for the reader.

We removed the technical aspects from this section for clarity.

Section 2.3: Confirmation of a pathogenic role for autoantibodies and autoreactive T cells

This seems to introduce the following sections, it should not be proposed at the same “Level” (2.3, 2.4, 2.5.) of the next chapters.

The title “Confirmation of a pathogenic role for autoantibodies and autoreactive T cells” has been removed and the text integrated under section 2.2.

Section 3.2.1 Anti-Yo autoantibody associated PCD

Section 3.2.1 arrives after section 2.5. Starting from this section the Authors introduces a new formal convention, the sub-section "Pathomechanism" that was never seen before.

Each subsection of autoimmune cerebellitis includes a general introduction, discussion of pathomechanism, and potential therapeutic approaches. This is now clarified under 3.

After section 3.2.2. there is a new 2.5 section.

The enumeration of the sections has been redone.

Minor issues

Abstract: Line 7-9: It is hard to summarise the role of cerebellum and the neurological implications in two sentences, the mentioned “predictive fashion” does not describe the much more complex and still debated role of the cerebellum, the Authors should try to stick with a stricter neurophysiological interpretation.

We have rephrased the abstract.

Line 46-48: The mechanisms underlying the cell trafficking is not mentioned while it could better introduce the reader into the topic.

We now expanded the section regarding cell trafficking.

Figure 2 (b): The disruption of the BBB organisation in inflammatory conditions is not immediately apparent. The differences between the “Healthy” and “Inflamed” side should be better highlighted in the figure.

We introduced different colors to differentiate between the healthy and inflamed state.

Line 109-110: In this sentence, the impact of stress on BBB integrity is compared to the effects of infections and brain injuries. Such a strong statement would benefit of a specific citation.

We now provide citation for each of the triggering events.

Line 112-113: Since the autoimmune response seems to be one of the main topic in this review it may deserve more attention. The passage from an immune to an autoimmune response is described without mentioning the mechanisms underlying this transition.

We now added text addressing pathways leading to autoimmune responses.

The readability of the first chapter benefits from the presence of schematic figures. The second part of the Manuscript deals with even more complex mechanisms but lacks figure, its readability would greatly improve if the Authors added graphic schemes here too.

A figure depicting the different forms and pathogeneses of autoimmune cerebellar diseases has been added.

Reviewer 3 Report

Dear Authors

This is an excellent review paper. I suggest just including a table with the latest antibodies related to autoimune ataxias ( septin-5, Homer3 etc) 

Author Response

I suggest just including a table with the latest antibodies related to autoimune ataxias ( septin-5, Homer3 etc) 

We added a table accordingly.